# Building the Momentum for A Stronger Pharmaceutical System in Africa

**DOI:** 10.3390/ijerph19063313

**Published:** 2022-03-11

**Authors:** Silvia Ussai, Caterina Chillotti, Erminia Stochino, Arianna Deidda, Giovanni Ambu, Lorenzo Anania, Alberto Boccalini, Flavia Colombo, Alessandra Ferrari, Daniele Pala, Enrica Puddu, Giulia Rapallo, Marco Pistis

**Affiliations:** 1Clinical Pharmacology and Toxicology, University of Cagliari, 09124 Cagliari, Italy; g.ambu3@studenti.unica.it (G.A.); l.anania@studenti.unica.it (L.A.); a.boccalini@studenti.unica.it (A.B.); colombo.flavia@hotmail.com (F.C.); a.ferrari1@studenti.unica.it (A.F.); d.pala2@studenti.unica.it (D.P.); e.puddu10@studenti.unica.it (E.P.); g.rapallo@studenti.unica.it (G.R.); 2Department of Biomedical Sciences, Section of Neuroscience and Clinical Pharmacology, University of Cagliari, 09124 Cagliari, Italy; caterinachillotti@aoucagliari.it (C.C.); e.stochino@hotmail.it (E.S.); arideidda@hotmail.it (A.D.); mpistis@unica.it (M.P.)

**Keywords:** COVID-19, Africa, pharmaceutical system strengthening

## Abstract

Despite impressive progress, nearly two billion people worldwide have no access to essential medicines. The COVID-19 pandemic revealed Africa’s vulnerability due to its reliance on imports for most vaccines, medicines, and other health product needs. The vaccine manufacturing is complex and requires massive financial investments, with global, regional, and national regulatory structures introducing consistent and urgent reforms to assure the quality and safety of medicines. In 2020, there were approximately 600 pharmaceutical manufacturers in Africa, 80% of which were concentrated in eight countries: Egypt, Algeria, Morocco, Tunisia, Nigeria, Ghana, Kenya, and South Africa. Only 4 countries had more than 50 manufacturers, while 22 countries had no local production. Out of the 600, around 25% were multinational companies. Africa is equally affected by modest scaled capacities substantially engaging in packaging and labelling, and occasionally fill and finish steps, facing criticalities in terms of solvent domestic markets. This article discusses the challenges in the development of a local pharmaceutical manufacturing in Africa and reflects on the importance of the momentum for strengthening the local medical production capacity in the continent as a critical opportunity for advancing universal health coverage (UHC).

## 1. Background

“Nobody is safe until everybody is safe” has been a common mantra amid the pandemic, referring to the importance of the global collaboration to contain the spreading of the SARS-CoV-19 and including, among others, having equitable access to the vaccinations [1].

With one quarter of all global health expenditure on medicines, pharmaceutical policies are crucial to promoting health and achieving sustainable development [2].

Despite impressive progress, nearly two billion people worldwide have no access to essential medicines [3]. While COVID-19 Vaccines Global Access (COVAX)—the global initiative led by Global Alliance for Vaccines and Immunisation (Gavi), the Vaccine Alliance, the World Health Organization, and the Coalition for Epidemic Preparedness Innovations (CEPI), alongside UNICEF and aiming to ensure fair access to COVID-19 vaccines worldwide—has sent out country allocations of 910 million doses delivered as of 30 December 2021 to all 190 COVAX countries, at the current vaccination rate it would take 5 years to cover 75% of the world population [4]. The number of COVID-19 vaccination doses administered in low-income countries stands at 0.2 percent of the population, compared to 16.7 per cent in middle-income countries and 48.7 per cent in high-income ones. Africa is largely relying on Oxford/AstraZeneca doses from COVAX coming through the Serum Institute of India, with only Egypt, Guinea, Morocco, and Seychelles having provided the population with any of the vaccines available on the market [5]. With the rapid rise of the highly transmissible Delta variant, the spread of Omicron variant, which was revealed to be between two and four times more contagious than Delta, as well as the uncertainty about the duration of immunity provided by vaccines, high income countries are considering providing a third vaccine booster shot to their populations, prioritizing immunocompromised groups. At the EU level, the Commission exercised its option to purchase an additional 150 million Moderna doses, on top of another 1.8 billion doses from BioNTech/Pfizer, projected to arrive at the end of 2021. The U.K. has already purchased another 60 million doses of the BioNTech/Pfizer vaccine for its booster program. Germany—with Omicron accounting for 73.3% of cases nationwide—is also planning on purchasing 204 million vaccines for 2022, according to Reuters. At the same time, Africa is facing a “vaccine desert”, due to serious delays and shortages of vaccine supplies, leading to insufficient doses to continue the vaccination [6]. The reported disparities in the access to COVID-19 vaccines have financial consequences as well: a study by RAND projected that the unequal global and local distribution of COVID-19 vaccines could cost the world economy up to $1.2 trillion [7]. 

Recently, the COVAX supply mechanism itself has been questioned in terms of ethics, with a targeted distribution based on the burden of COVID-19 in the country and its health system capacity only foreseen in the second phase of the allocations, failing to recognize the significant differences in health outcomes and access to vaccines between countries (according to ‘The Fair Priority Model’) [8]. Furthermore, the buyer consolidation in the vaccines market, with Gavi financing >90% of market by volume and UNICEF Supply Division (SD) procuring approximately two-thirds of open market valuation, today solves a market failure, not a product failure. Among others, by 2030, up to nine African countries, which are benefitting from financing mechanisms to de-risk the market, could transition from Gavi—including Nigeria and Kenya—leading to uncertainties about the demand dynamics [9].

The Africa Union is composed of 55 countries, each one with relevant differences in terms of health systems and pharmaceutical manufacturing capacity. This has impacted their overall response to the pandemic and is further reflected, among others, with the current vaccination rate. It is understood that the COVID-19 vaccine research provided a strong push towards the vaccine development, pharmaceutical, and medical products in general. The aim of this viewpoint is to present the unique momentum Africa—with the unprecedented interests from traditional and emerging aid donors, as well as the engagement of the private sector—is experiencing towards the achievement of sufficient capacity in responding to its essential medical needs.

## 2. State of Art in the Domestic Pharmaceutical Production in Africa

In 2020, there were approximately 600 pharmaceutical manufacturers in Africa, 80% of which were concentrated in eight countries: Egypt, Algeria, Morocco, Tunisia, Nigeria, Ghana, Kenya, and South Africa. Only 4 countries had more than 50 manufacturers, while 22 countries had no local production. Out of the 600, around 25% were multinational companies [10]. 

According to a recent study commissioned by the UK Government, there are ten known existing local vaccine value chain players in Africa (mostly concentrated in North Africa, South Africa, and Nigeria), primarily producing limited volumes and/or downstream value chain steps [11].

Encouraging steps have been taken to ensure country-level production of vaccines amid the COVID-19 pandemic, including the European Commission €1 billion package of investment in vaccines, medicines, and health products for Africa [12].

In November 2020 the Russian Direct Investment Fund Morocco (RDIF) signed a deal with Moroccan pharmaceutical manufacturer Galenica to produce the Russian COVID-19 vaccine locally. Similarly, in 2021 the Aspen Pharmacare started the production of Johnson & Johnson COVID-19 vaccines in South Africa [13]. The company BioNTech has recently signed an agreement with the Rwandan government and Institut Pasteur de Dakar in Senegal on the construction of the first mRNA vaccine manufacturing facility in Africa starting in mid-2022.

Nevertheless, these virtuous business cases are clearly insufficient to address the population needs in Africa [14].

## 3. Challenges and Opportunities for a Pharmaceutical Local Production in Africa

The vaccine manufacturing is complex and requires massive financial investments, with global, regional, and national regulatory structures introducing consistent and urgent reforms to assure the quality and safety of medicines. The disruption in the vaccines rollout recently affecting Africa, with millions of COVID-19 doses expiring before their administration (due to the vaccines’ short shelf life combined with the logistic/operational challenges), revealed all the dysfunctionalities of the African pharmaceutical value supply chain. Among others, Malawi destroyed almost 20,000 doses of the AstraZeneca (AZ) vaccine, while South Sudan announced it would destroy 59,000 doses [15,16]. 

According to the UNECA, Africa covers the vast majority of its needs through imports, mainly from Europe (51.5%), India (19.3%), Switzerland (7.7%), China (5.2%), United States (4.3%), and United Kingdom (3.3%) [17]. Currently, Africa’s reliance on vaccine imports is caused by various interrelated constraints impeding local production. These include, inter alia: (i) no clear agenda or co-ordination across efforts, (ii) weak regulatory and supply chain value environments (e.g., fragmented offerings, high distribution costs, unclear demand preventing to define the return of investment), (iii) bifurcated market demand dynamics (Gavi vs. non-Gavi), (iv) restricted access to finance, and (v) limited local talents [18].

Strengthening domestic pharmaceutical systems technically involves strengthening the building blocks of health systems, including policy, laws, governance; regulatory systems; innovation, research and development, manufacturing, and trade; financing; human resources and information.

Africa is equally affected by modest scaled capacities substantially engaging in packaging and labelling, and occasionally fill and finish steps, facing criticalities in term of solvent domestic markets. For instance, Tunisia’s 39 manufacturers, operating primarily as joint ventures with international firms, covered 52% of the domestic demand in 2019 and targeted 62% for 2023 (projections before the COVID crisis). Tunisian exportations to other African countries remain constrained by infrastructure allowing for direct air connections, maritime routes, or business events. Similar patterns are found in Morocco and Algeria. [19].

At the global level, evidence suggests that countries which have been transforming their norms on Intellectual Properties (IP), have also been undergoing radical regulatory restructuring of their pharmaceutical markets in terms of product registry and regulation [20]. In this light, the World Trade Organization (WTO) Agreement on Trade-related Intellectual Property Rights (TRIPS), which sets the minimum standard of protection of, inter alia, copyright, trademarks, geographical indications, industrial designs, and trade secrets, also foresees a certain level of flexibilities, particularly in case of health emergencies. India and South Africa, co-sponsored by a large number of developing countries, submitted an initial proposal for a temporary waiver in response to COVID-19 in October 2020, followed by a revised proposal in May 2021, which continues to divide opinion. While the World Health Organization (WHO) and many human right organizations revealed to be in favor of this option, at least in the short term, several WTO members (among others, Australia, Japan, Norway, Singapore, South Korea, Switzerland, and Taiwan) hold reservations about starting text-based negotiations on a temporary TRIPS waiver. The European Member States position has also been reportedly reserved, due to concerns about the impact on pharmaceutical innovation. In addition, international mergers and acquisitions activities over the last decades have resulted in unprecedented consolidation across all levels of the global pharmaceutical supply chain, with fewer suppliers increasing competitive pressure for the public procurement programs and for local producers. At the same time, second-tier suppliers have flooded the international pharmaceutical market with products which may not meet quality standards [21]. Developing countries, including Africa, therefore face a double-edged regulatory challenge: harmonizing their IP rights norms to global standards and working to upgrade and implement quality standards for local populations. 

Introducing biomedical products on a country’s market requires strict administrative authorization, best delivered by a Stringent Regulatory Authority (SRA), for quality assurance and fraud prevention. 

Except for few countries, reliable SRAs are still missing in Africa despite regional initiatives in that direction. An estimated 1 in 10 medical products in low- and middle-income countries is substandard or falsified, according to the World Health Organization [22]. Another factor influencing the increase in the size of pharma emerging markets is the up-selling of local generic drugs. It is equally important to distinguish between different levels of production of generic medicines: manufacturing raw materials from basic chemical and biological substances, synthesis of active pharmaceutical ingredients, producing dosages from the active pharmaceutical ingredients, and packaging and labelling finished products. Local manufacturers face a cost disadvantage since finished medicines can be imported duty free, while active pharmaceutical ingredients (APIs), which account for 60% or more of the final cost of the product, carry a duty. An additional layer of complexity invests vaccines, as second-generation vaccine manufacturers must develop their own production processes (including additional clinical trials), due to lot-to-lot testing for specific efficacy and safety that cannot be determined by simple bioequivalence, overall leading to considerable costs [23].

In this light, the strengthening of the local vaccine manufacturing capacity represents an opportunity for advancing universal health coverage (UHC) in Africa and ensure equitable access to essential medicines, as the Sustainable Development Goals and associated targets explicitly refer to the need for affordable vaccines. 

## 4. Lessons Learnt from the COVID-19 Pandemic

There are unequivocal lessons learnt from the COVID-19 pandemic in Africa. These are, among others: (i) the urgency to strengthening regional and national regulatory systems and their governance through reliance mechanisms, open science, and regulatory harmonization; (ii) a clear demand for supporting the local manufacturing in the context of the Manufacturing Plan for Africa and the operationalization of the African Medicines Agency, which can be performed via a step-wise and diversified approach according to the technological maturity and the availability of infrastructure in the various partner countries; (iii) the equal importance to advance on vaccine products-related value chains mapping at country level, which could be envisaged in the regional economic commissions (RECs) multi-annual strategies. Indeed, continental and regional approaches under the auspices of the African Union through the RECs revealed to be critical in leu of the limited capacity of some states to manage the pandemic. There is also a clear need for the continent to establish affordable financing mechanisms, as well as establish a framework for collaborations with the aim to strengthen a coordinated informed buying system across different RECs.

## 5. Conclusions

With this manuscript, the authors acknowledge the opportunity to expedite and support local production of pharmaceuticals and the health industry in Africa.

The persistent inequity in access to vaccines and the emergence of the Omicron variant, as well as other COVID-19 variants reflects the need for more widespread regional base vaccine production, addressing crucial issues including, among others, the technology transfer as well as appropriate intellectual property reform on essential medicines and vaccines.

The current pandemic has helped to build the momentum for a strong pharmaceutical system in Africa, considering a new proposed strategy for continental health promotion, which will contribute building trust and confidence of the local communities [24]. This means a call for action from both supply and demand sides. Specifically, the local industrial development will play a critical role in financing new and expanding an already existing manufacturing capacity. The introduction of a quality assurance system throughout the supply process is also a critical element to address the plague upon the sub-standards and pharmaceutical falsification in Africa. On the health financing, the removal of tariff and non-tariff barriers on pharmaceutical products, as well as the introduction of trade facilitations (e.g., regional free trade areas) will contribute to creating a robust business environment and shape the market, particularly for GAVI transitioning countries. Finally, the execution of such momentum includes the introduction of several enabling factors, such as an improving regulation and governance of pharmaceutical products, with the adoption of coherent national policies providing incentives and promoting human capital development.

## Data Availability

Not applicable.

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
