# Peer review of "Building the Momentum for A Stronger Pharmaceutical System in Africa"

_ijerph, 2022, doi:10.3390/ijerph19063313_

Round 1

Reviewer 1 Report

This is an important and very current topic which is being debated widely. on the global stage and which needs to be discussed widely in peer reviewed articles. Nevertheless there are a number of significant issues that need to be addressed before this paper could be considered suitable for publication.

  • It is not clear if this article is about COVID19 vaccines or vaccine development more generally. This needs to be made much clearer. It can of ocurse be about both, but currently it jumps between the two topics somewhat indiscriminately and leaves the reader a little confused.
  • In the background, much of the content is out of date;  g % vaccinated, boosters given and importantly the advent of Omicron which actually isn’t mentioned until  the conclusion. There are a number of terms that are introduced but not explained at all to the reader (COVAX) (GAVI).
  • Africa is a large continent. Whilst there is some mention of the different production capacity in different countries, there is no sense from the paper of the very different manufacturing capability or heath systems issues facing different countries.
  • None of the issues touched on are discussed in any depth and no conclusion is drawn with regards to feasibility. The article touches on IP and indeed finishes by mentioning reform as a crucial issue but does not even mention the current TRIPS waiver debate or the role of the WTO. Neither do the authors incorporate any of the current arguments for and against local vaccine manufacture that are currently being debated across the world, or attempt to unpick the stance of the pharma industry versus the views of the WHO and many human rights organisations around the world. Infact the stance of the Pharma industry and the solutions it proposes, are not really discussed.  Further, where important points are touched on, they appear to be only mentioned briefly without any political context or opinion on whether or how they might be overcome  g. supply chains.

Where ‘recommendations’ are made, they are quite sweeping and agan without  context  (e.g. There is also a clear need for the continent to establish affordable financing mechanisms as well as establish a framework for collaborations with the aim to strengthen a coordinated informed buying system across different RECS.

  • The conclusion is very weak and leaves the reader wondering whether the authors believe it is feasible to promote vaccine production in Africa or not.
  • The paper needs to be edited by someone with English as a first language

In my view, the paper needs to be substantially re-written, and should rehearse the many political and practical arguments that are currently being played out with regards to vaccine manufacture in Africa, before reaching a clear conclusion on the potential way forward for the issue at hand.

Author Response

REVIEWER 1

This is an important and very current topic which is being debated widely. on the global stage and which needs to be discussed widely in peer reviewed articles.

We thank the reviewer for the positive considerations.

It is not clear if this article is about COVID19 vaccines or vaccine development more generally. This needs to be made much clearer. It can of course be about both, but currently it jumps between the two topics somewhat indiscriminately and leaves the reader a little confused.

It is understood that the COVID-19 pandemic provided a strong push towards the vaccine development in general. In this sense, the article starts by discussing the issues strictly related to COVAX and refers to the “vaccines” when contents are applicable to the general immunization environment. In any case, we have reviewed the paragraphs to harmonize the text and mitigate the discomfort perception of jumping from one topic to the other.

In the background, much of the content is out of date;  g % vaccinated, boosters given and importantly the advent of Omicron which actually isn’t mentioned until  the conclusion. There are a number of terms that are introduced but not explained at all to the reader (COVAX) (GAVI).

We agree with the reviewer about the issue of having some outdated data in the article. Indeed, this Opinion has been finalized in mid-December, with an epidemic situation continuously evolving, including Omicron isolation. This has limited our opportunity, among others, to discuss consolidated data on the variant. Epidemiological data have been now introduced in the text, along with clarification on GAVI and the COVID-19 Vaccines Global Access (COVAX).

Africa is a large continent. Whilst there is some mention of the different production capacity in different countries, there is no sense from the paper of the very different manufacturing capability or heath systems issues facing different countries.

Authors agree with this consideration and would like to thank the reviewer for having risen the issue.

Africa Union is composed by 55 Countries, each one with relevant differences in terms of health systems and pharmaceutical manufacturing capacity. This has of course impacted their overall response to the pandemic and is further reflected, among others, with the current vaccination rate. However, we would like to highlight the fact that our aim goes beyond providing a review of the pharmaceutical/vaccine capacity across the Continent. Starting from the acknowledgment that there are different levels of maturity in Africa, we would rather intend to highlight the need for the Continent to achieve a sufficient capacity to respond to their essential medical needs. We mentioned some examples in the text to capture common possible barriers and opportunity, although we are very conscious of the “context specifications”. This will also be made possible thanks to the support of traditional and non-conventional donors. Several authors contributing to this paper have personally worked with international organizations and agencies engaged in strengthening the local production of vaccines and medicines in Africa, including COVID-19 vaccine. Following the reviewer’s input, we have tried to clarify those aspects in the text.

None of the issues touched on are discussed in any depth and no conclusion is drawn with regards to feasibility. The article touches on IP and indeed finishes by mentioning reform as a crucial issue but does not even mention the current TRIPS waiver debate or the role of the WTO. Neither do the authors incorporate any of the current arguments for and against local vaccine manufacture that are currently being debated across the world, or attempt to unpick the stance of the pharma industry versus the views of the WHO and many human rights organisations around the world. Infact the stance of the Pharma industry and the solutions it proposes, are not really discussed.  Further, where important points are touched on, they appear to be only mentioned briefly without any political context or opinion on whether or how they might be overcome  g. supply chains.

We thank the reviewer for the interesting inputs provided.

We have included in the text a paragraph on TRIPS waiver debate or the role of the WTO. We have also incorporated arguments for and against local vaccine manufacture that are currently being debated across the world, or attempt to unpick the stance of the pharma industry versus the views of the WHO and many human rights organisations around the world.

In terms of providing political context, we appreciate the suggestion. However, although we believe it is out of the scope of the current article, we are committed to submit another paper including the political implications of the topic, also in light of the recent visit from the EC President in Senegal.

Where ‘recommendations’ are made, they are quite sweeping and again without context  (e.g. There is also a clear need for the continent to establish affordable financing mechanisms as well as establish a framework for collaborations with the aim to strengthen a coordinated informed buying system across different RECS.

Authors have further expanded those concepts and linked them also with a broader discussion on sustainability. The role of RECs is here important: as we have mentioned previously, we did not meant to discuss specifications from each country; however similarities can emerge from countries belonging to the same REC.

The conclusion is very weak and leaves the reader wondering whether the authors believe it is feasible to promote vaccine production in Africa or not.

We agree with the reviewer’s comment. As researchers, we aim to highlight the fact that, for the very first time ever, supporting the vaccine and pharmaceutical capacity in Africa has become a priority in the global health agenda. With 1B€ donation from the European Commission, this is a critical time to build the momentum. Our position has been clarified in the conclusion.

The paper needs to be edited by someone with English as a first language

The paper has been edited by an English native speaker.

Reviewer 2 Report

This manuscript needs to provide supporting evidence or even preliminary data. Contents and composition of the manuscript needs to be improved for proper research content. 

Author Response

REVIEWER 2

On behalf of my co-authors, I would like to thank the reviewer for the article assessment. The Opinion has been extensively revised and greatly improved. We believe it is now feasible for publication.

Round 2

Reviewer 2 Report

The conclusions need to be more specific how the pandemic helps the momentum for strong pharmaceutical system in Africa and how to execute such momentum.

Author Response

The conclusions need to be more specific how the pandemic helps the momentum for strong pharmaceutical system in Africa and how to execute such momentum.

On behalf of my co-authors, I would like to thank the reviewer for the additional comment provided. We believe that overall the recommendations are contributing to improve the quality of this article.

The conclusion paragraph has been further expanded to discuss how the pandemic has impacted the momentum for the pharmaceutical support in Africa as well as several points have been included about the possible operationalization of this momentum in the continent (both from supply and demand side).

We believe the article complies with the suggestions and is eligible for the publication.